materials science

graphene oxide, montmorillonite, liquid crystals, aerogel

**Author for correspondence:**
Zhihong Yang
e-mail: yzhh05@126.com

This article has been edited by the Royal Society of Chemistry, including the commissioning, peer review process and editorial aspects up to the point of acceptance.

# Clay – graphene oxide liquid crystals and their aerogels: synthesis, characterization and properties

Sisi Ye, Zhihong Yang, Jianmei Xu, Zehao Shang and Jing Xie

Faculty of Materials Science and Chemistry, China University of Geosciences, Wuhan 430074, People's Republic of China

SY, 0000-0002-3926-0331; ZY, 0000-0002-1884-5429

The dispersions containing two kinds of layered solids—graphene oxide (GO) and exfoliated montmorillonite (MMT) were mainly prepared, following which the binary aerogels were synthesized. The results indicate that the formation of liquid crystals (LCs) occurs at lower GO concentration and the birefringence becomes stronger when MMT is introduced into GO dispersion. Sol–gel transition forms in the binary suspensions with different mass fractions of MMT and GO. LCs with highly ordered alignment are observed in the gel and the fraction of LCs obviously increases with the increase in GO concentration. Moreover, the birefringence is observed in MMT–GO binary aerogels with the interconnected three-dimensional porous network, which is attributed to the ordered arrangement of MMT and GO nanosheets in pore walls. Among the aerogels with different MMT/GO ratios, the samples at the ratio of 10:1 show better adsorption capacity and removal percentage of cationic and anionic dyes.

## 1. Introduction

The mineral clay and graphite are structurally organized as two-dimensional solids and their monolayer exfoliation has aroused tremendous interest in the world. Their monolayers with a plate-like shape are highly anisotropic colloidal particles with high aspect ratios (ratio of width to thickness) and low thicknesses less than a nanometre [1]. Additionally, clay and graphene oxide (GO) (the oxygenated monolayer of graphite) possess good solubility and form stable colloidal suspensions in polar solvent. Based on the high aspect ratio and good solubility, it is possible for these two-dimensional particles to form liquid crystals (LCs) through spontaneous self-assembling of the building blocks [2], driven by

the competition between packing and orientation entropies [3]. LC formation offers a versatile route to assemble the macroscopic-oriented materials [4], such as orientational nanocomposites, porous materials and functional soft materials with anisotropy [5].

Montmorillonite (MMT), one of the typical layered clay minerals, is of great industrial value and employed in many commercial fields due to its natural abundance, intercalative capacity, environment-friendly nature, high aspect ratio, low cost and large surface area [6,7]. It has the capability for soaking up a large amount of water with dramatic osmotic swelling. Hence, it is spontaneously exfoliated into aluminosilicate monolayers and forms stable colloidal dispersion without any physico-chemical treatment or exfoliating agent [8]. Moreover, it is interesting that for the dispersion with high clay concentration, the sol/gel transition occurs and LCs can be observed in the gel. As early as in 1956, Emerson [9] reported that a large banded texture presented in swollen MMT hydrogel under polarizing microscope, identifying the extension of LCs over macroscopic distances. Gabriel *et al*. [10] observed that stable optical birefringence existed in the concentrated bentonite gel suspensions (more than 24 mg ml$^{-1}$), and only transient flow birefringence was detected at concentrations slightly below 24 mg ml$^{-1}$. Therefore, with increasing the concentration, clay suspensions exhibit the transition from an isotropic liquid to an isotropic gel and then to a birefringent gel [11]. These reports indicate that liquid crystallines occur in the gel suspension with high MMT concentration [12].

Graphite is often chemically exfoliated to synthesize GO that is the oxygenated form of a monolayer graphene [3,4]. Owing to the strong mechanical strength, chemical functionalization capability and extremely large surface, GO shows potential applications in many fields such as optical nonlinearity [13], water purification [14], polymer composites [15], coating [16], lens [17], flexible rechargeable battery electrode [18], etc. GO exhibits good dispersion in water because its surface is rich in hydrophilic oxygen-containing groups such as epoxy, hydroxyl and carbonyl groups. The well-dispersible GO can form highly oriented LCs in aqueous dispersion at very low concentration (0.25 mg ml$^{-1}$) [3]. Through self-organization or self-assembly of nanoscale building blocks [3,19], GO LCs have been employed to prepare macroscopic-oriented materials including one-dimensional nanomaterials [20], two-dimensional films and three-dimensional ordered foams with fantastic performance [21].

Considering the similar two-dimensional structures of GO and exfoliated clay, the combination of GO and MMT has aroused new interest. It has been identified that the mixture of GO and MMT generates synergy effect and holds great potential in many fields [14,22,23]. But there is one controversial issue about the nanoparticle arrangement of GO–MMT hybrid in the literature. Fang *et al*. [20] found that LCs alignment occurred in GO–MMT hybrid suspension and hence the wet-spun fibres with oriented microstructure were prepared from GO–MMT LCs. However, Mangadlao *et al*. [24] believed that GO–MMT suspension exhibited 'house of cards' network and the aerogels with disordered structure were obtained from the hybrid solution. Other reports [25,26] involved with GO–MMT hybrid materials are hardly focused on the texture of the dispersion though the nanosheet arrangement affects the properties of the prepared macroscopic materials.

In our study, the texture of MMT–GO binary dispersion was deeply investigated using a polarized-light optical microscope (POM). Additionally, the aerogels from MMT–GO solution were synthesized based on the combination of LCs self-assembly and ice-templating strategy, and their microstructure and absorption property were also investigated.

# 2. Experimental section

## 2.1. Materials

Graphite was obtained from Sinopharm Chemical Reagent Co. Ltd. Na-montmorillonite was donated by Zhejiang Sanding Technology Co. Ltd. Methylene blue (MB) and methyl orange (MO) were purchased from Shanghai Shanpu Chemical Co. Ltd and Tianjin Damao Chemical Reagent Factory, respectively.

## 2.2. Preparation of GO–MMT hybrid dispersion

GO was prepared from the graphite powders using Hummer's method [27,28]. High-speed centrifugation together with dialysis was applied to remove the acidic or ionic impurities from the as prepared GO. The bulky MMT was vigorously stirred in deionized water for one week to obtain exfoliated MMT solution. The prepared MMT and GO aqueous solutions were centrifuged at 4000 r.p.m. for 30 min to remove the

unexfoliated particles. Then MMT suspension and GO dispersion were mixed through stirring for 60 min and subsequently sonicated for 30 min to obtain homogeneous GO–MMT hybrid dispersion.

## 2.3. Preparation of GO–MMT aerogel composites

Desired amount of MMT and GO suspensions are homogeneously mixed to obtain hybrid sols with different mass fractions of MMT and GO: 0.8 wt% GO–2 wt% MMT, 0.5 wt% GO–2.5 wt% MMT, 0.3 wt% GO–3 wt% MMT and 0.1 wt% GO–4 wt% MMT, which are corresponding to GO/MMT ratios of 1 : 2.5, 1 : 5, 1 : 10 and 1 : 40, and the resulting composites were denoted as $M_{2.5}G$, $M_5G$, $M_{10}G$, $M_{40}G$, respectively. Additionally, the pure clay sol with 5 wt% MMT was prepared and denoted as $MG_0$. Then the mixture sols were poured into the homemade cylinder-shaped mould of aluminium foil (diameter of 20 mm and height of 45 mm). After sol–gel transition was finished, the gels were frozen in a liquid nitrogen bath and subsequently freeze-dried for 3–4 days to obtain aerogel composites.

## 2.4. Adsorption experiment

Adsorption tests of MB and MO dyes onto the aerogels were conducted at room temperature. MB and MO solutions with the concentrations of 150 and 500 mg $l^{-1}$, respectively, were prepared. Then aerogel samples weighing 0.015 and 0.15 g were added in 50 ml of MB and MO solution, respectively. The mixture was stirred gently for 2 h, and subsequently the absorbency of the solution was measured by a UV–vis spectrophotometer at the wavelength of 664 nm (MB solution) and 464 nm (MO solution). The removal amount of cationic or anionic dye per unit mass of adsorbent ($q_e$, mg $g^{-1}$) was determined according to the following equation:

$$q_e = \frac{(C_0 - C_e)V}{M},$$

where $C_0$ is the initial concentration (mg $l^{-1}$), $C_e$ is the equilibrium concentration (mg $l^{-1}$), $V$ is the liquid volume (l) and $M$ is the dosage of the solid adsorbent (g).

## 2.5. Characterization

To investigate LCs of GO–MMT dispersion, the dispersion was filled in a capillary or trapped on a glass slide whose edges were sealed. Then the arrangement of GO–MMT sheets was observed by a POM with OLYMPUS OA01.

Atomic force microscopy (AFM) images of GO sheets and MMT nanoplatelets on mica substrates were determined using an NSK SPI3800. X-ray diffractometer (XRD) pattern was conducted at a scanning rate of $10° min^{-1}$ using Bruker D8-ADVANCE. Fourier-transform infrared spectroscopy (FTIR) was carried out using KBr pressed disc technique with a Thermo Nicolet6700 spectrophotometer. Scanning electronic microscope (SEM) was performed using Hitachi SU-70 and Philips CM12 equipment. Zeta potential of the solution was measured on a Zetasizer Nano ZS90 system.

# 3. Results and discussion

Figure 1 shows that both GO and MMT sheets exhibit irregular polygonal shapes. The lateral size of GO nanoplatelets is in the range of 3–30 μm and the thickness is around 1.2 nm, demonstrating the single-layer state of GO and high aspect ratio over 2500. The large lateral size and high aspect ratio of GO are beneficial to the spontaneous formation of lyotropic LCs at low concentration. The diameter of MMT nanoplatelet is in the range of 0.5–2 μm and the thickness is about 2.76 nm with the aspect ratio less than 725, much lower than that of GO flakes. According to the monolayered silica sheets with a theoretical thickness of 0.96 nm [29], MMT nanosheets are composed of two to three monolayers.

The effect of MMT addition on LC formation of GO dispersion is investigated, as shown in electronic supplementary material, figures S1–S3. The nematic LCs are observed in GO and GO–MMT suspension. Suspensions with GO concentration of 0.1 wt% display small LCs with weak birefringence (electronic supplementary material, figure S1). With increasing MMT addition, the optical textures become more compact and the birefringence domains gradually connect with each other, implying that the fraction of LCs increases. Moreover, various interference colours are observed in the dispersions with different mass fractions of MMT–GO hybrids. Especially for the suspension with high GO concentration at

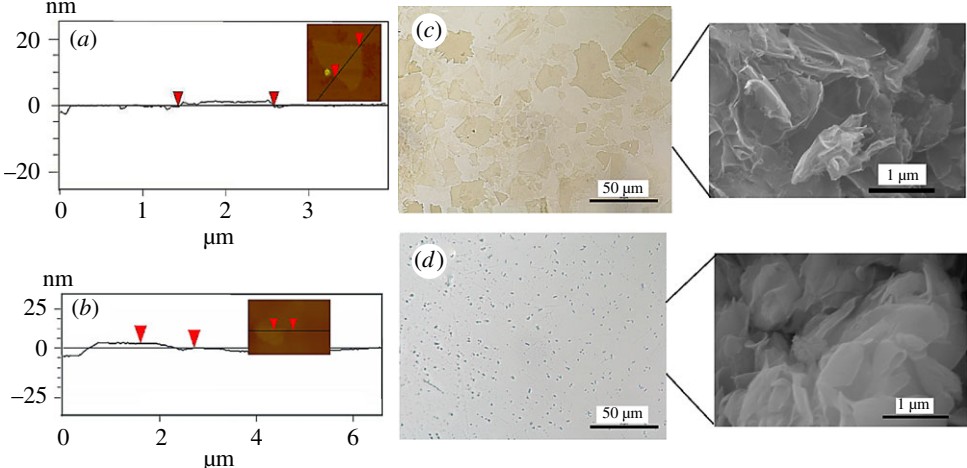

**Figure 1.** AFM images and heights of GO (*a*) and MMT platelets (*b*). Optical microscope images and SEM images of GO (*c*) and MMT (*d*).

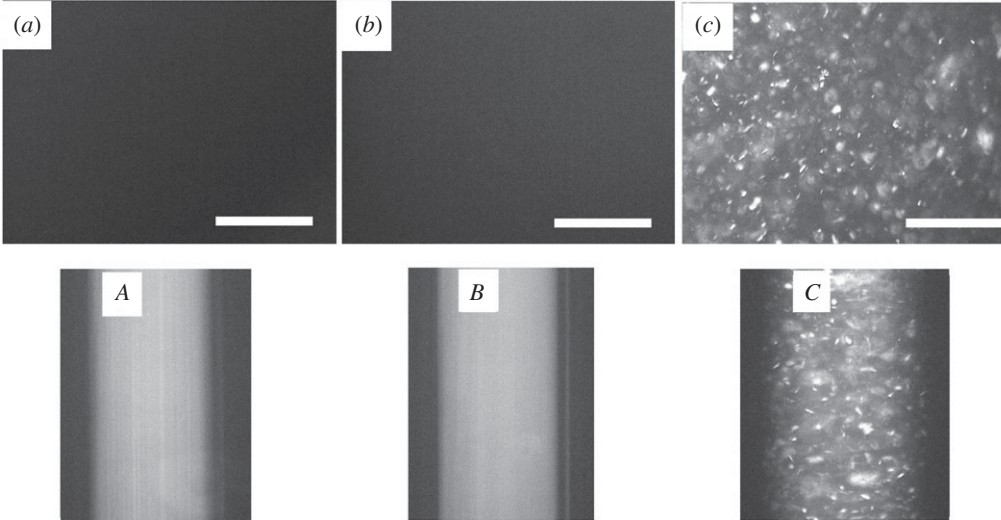

**Figure 2.** POM images of the suspension on a glass slide with 0.02% GO (*a*), 0.25% MMT (*b*), 0.02%GO−0.25%MMT hybrids (*c*). POM images of the corresponding suspensions in a capillary (*A−C*). Scale bar, 400 μm.

0.3 and 0.5 wt% (electronic supplementary material, figures S2 and S3), birefringence colours change gradually from white to light yellow, orange and red in order, indicating that the birefringence becomes stronger and the amounts of LCs increase with the increase in MMT concentration [10,30]. Therefore, it can be concluded that the introduction of MMT nanosheets enhances LCs formation of GO suspension.

As shown in figure 2, the single suspension containing 0.25 wt% MMT or 0.02 wt% GO displays isotropic state without birefringence. It is interesting that many LCs domains occur in the binary dispersion containing the same concentration of MMT and GO. So, from the results of POM images as shown above, it can be seen that the combination of MMT and GO nanoparticles shows a good synergetic effect on LCs formation.

MMT−GO suspension shows the evolution of LCs phases as a comparison with their native counterparts, which is mainly attributed to the confining effect of GO particles [20]. GO flakes are negatively charged with a zeta potential of −62.4 mV (figure 3*a*) due to the surface functional groups such as carboxyl and hydroxyl [3]. Also, the basal planes of MMT are permanently negatively charged with a zeta potential of −44.5 mV because of cation replacement in the octahedral sheet ($Mg^{2+}$ for $Al^{3+}$) and in the tetrahedral sheet ($Al^{3+}$ for $Si^{4+}$) [12,31]. After MMT and GO are mixed, the negatively charged hybrids show zeta potential values between −62.4 and −44.5 mV. So MMT and GO nanosheets attempt to repel each other due to the electrostatic interaction, which determines the good dispersive stability of MMT and GO sheets in aqueous solutions. GO flakes show stronger electrostatic repulsion than MMT sheets due to the lower zeta potential, indicating that there exists the large interlayer space between the parallel GO flakes and it

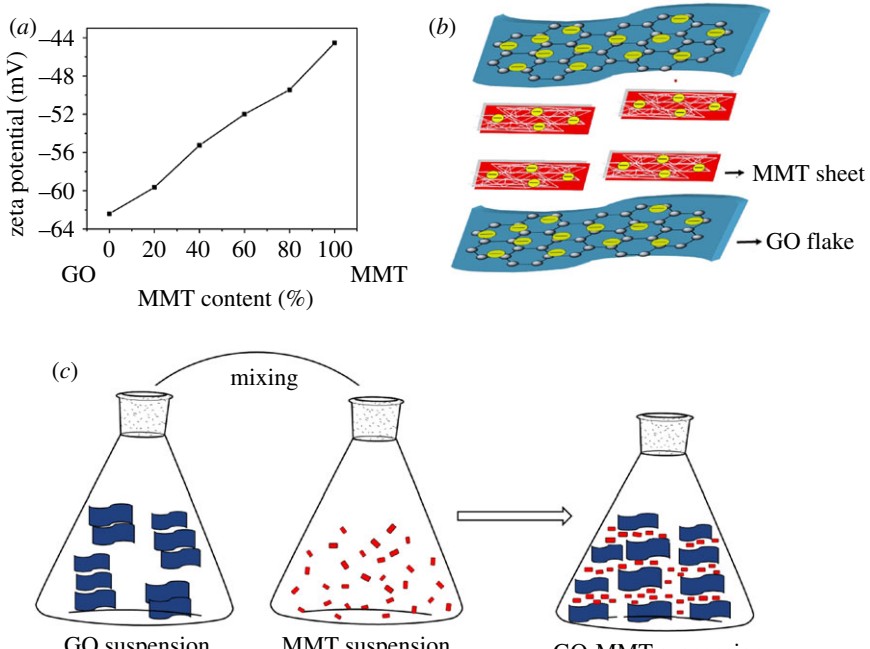

**Figure 3.** Zeta potential of MMT, GO and their hybrid suspension (*a*). Diagram of MMT intercalation between GO flakes (*b*). Diagram of LCs formation of GO–MMT hybrid suspension (*c*).

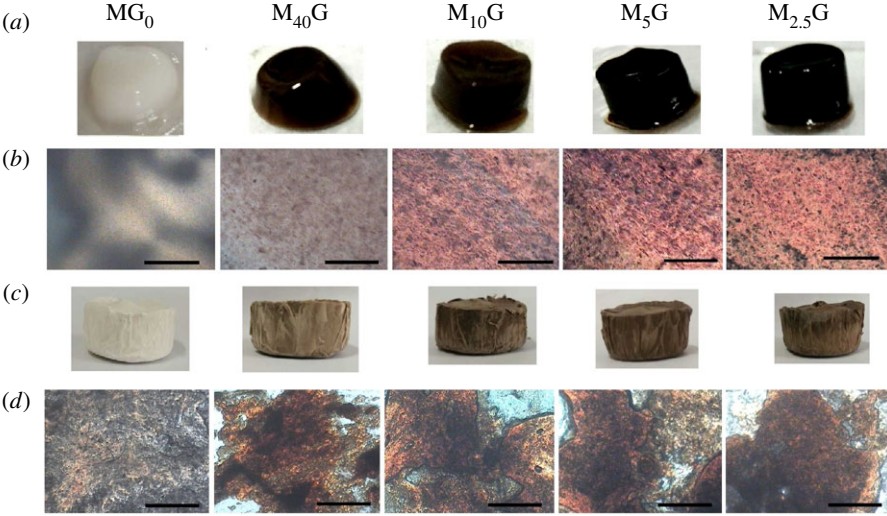

**Figure 4.** Digital photos of the gels (*a*). POM images of the corresponding gels (*b*). Digital photos of the corresponding aerogels (*c*). POM images of the corresponding aerogels (*d*). All the images from left to right in order: $MG_0$, $M_{40}G$, $M_{10}G$, $M_5G$, $M_{2.5}G$. Scale bar, 400 μm.

is available for the intercalation of small MMT sheets [32]. As shown in figure 3*b*, MMT nanosheets are sandwiched and confined between GO flakes, so the face-to-face arrangement of MMT platelets occurs due to an excluded volume effect in the limited space [33]. The aligned arrangements of exfoliated MMT sheets cause the larger nematic domains, revealing the templating role of GO LCs in MMT–GO hybrid suspensions [20] (figure 3*c*). Therefore, compared with GO and MMT aqueous solutions, their combination shows strong birefringence and synergistic LCs formation.

The sols with proper GO–MMT mass fraction could be converted into a gel due to the electrostatic interactions of the aligned nanosheets. Figure 4*a*,*b* shows the photos of GO–MMT gels and their corresponding POM images. The nematic schlieren textures occur in the gel with 5 wt% MMT, which is in agreement with the reports that LCs phases exist in clay gel with high concentration. When GO was introduced into MMT dispersion, the obviously parallel-banded textures are observed between crossed

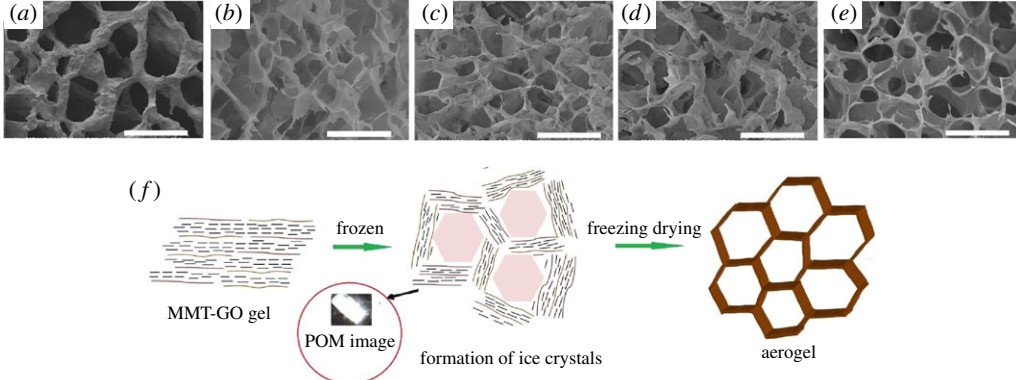

**Figure 5.** SEM image of the aerogel samples (a) MG$_0$, (b) M$_{40}$G, (c) M$_{10}$G, (d) M$_5$G, (e) M$_{2.5}$G. Schematic of the formation mechanism of hexagonal shape pores through the growth of ice crystals (f).

polarizers, indicating that the LCs evolve to more ordered mesophases like a lamellar phase. It is known that there are two different issues involved with the organization of clay gel and GO–MMT gel: 'house of cards' arrangement and 'nematic organization'. Some researchers believed that the attractive interactions between natively charged edges of clay platelets and faces of clay/GO sheets caused 'house of cards' tridimensional network [24,34], while others reported that the electrostatic repulsions between the sheet faces induced the ordered parallel array [10,12]. In this study, ordered LCs observed in the gels and aerogels are in keeping with the view of parallel arrangement not 'house of cards' organization.

LCs self-assembly and freeze-drying technology can give birth to macroscopically assembled materials [35] and hence they were employed to prepare aerogels in this study. Figure 4c shows the aerogels prepared from the LCs gels that have been frozen in liquid nitrogen followed by freeze-drying. POM images of the corresponding aerogels show that the birefringence is observed and the materials are also anisotropic (figure 4d), indicating that the oriented materials consisting of MMT or GO–MMT are synthesized.

SEM was employed to observe the architecture of the as-obtained aerogels (figure 5). A well-defined and continuous interconnected three-dimensional porous network is detected in all aerogels. The pore shape is attributed to the role of ice crystal template when the gel samples are frozen in a liquid nitrogen bath [36]. As shown in figure 5f, the growth of hexagonal ice crystals pushes away the sheets and hence the ordered nanosheets in the LCs are entrapped between neighbouring ice crystals [37], forming the pore wall with the parallel alignment of MMT–GO lamellars. Like the templating role of GO LCs in the solution, GO sheets provide the wrinkled skeletons of the pore walls by randomly overlapping or restacking one another [38], and MMT nanosheets are confined and orderly stacked between GO flakes. So, the pore walls of the as-prepared aerogels exhibit birefringence between crossed polarizers (figure 5d) due to the highly ordered stacking structure. Therefore, the formation of oriented interconnected three-dimensional porous aerogels is attributed to the combined effect of original GO–MMT LCs and ice crystal formation [35].

As shown in figure 6a, GO prepared through Hummer's method exhibits a diffraction peak at 11.5°, corresponding to the layer-to-layer stacking distance of 0.76 nm, which is much larger than the theoretical interlayer space of graphite [39]. MMT mineral displays a (001) peak at 7.7°, corresponding to a basal spacing of 1.15 nm. In figure 6b, all aerogel samples show a small peak at 7.7°, almost the same as that of MMT mineral, indicating that some sheets of swollen MMT get closer to each other and restack due to the electrostatic and van der Waals interactions upon freeze-drying [38]. But these characteristic peaks are much weakened and widened in the aerogels as a comparison with MMT mineral, which is attributed to the much low crystallinity. In addition, the characteristic peak of GO does not appear in the pattern of the aerogel samples, clearly indicating that GO sheets are fully exfoliated into individual sheets. So, XRD results further identify that GO flakes are separated from each other by the intercalation of ordered MMT sheets.

In figure 6c, FTIR pattern of MMT shows the bands at 362 cm$^{-1}$ (–OH stretch of structural hydroxyl), 3456 cm$^{-1}$ (–OH stretch of absorbed water), 1643 cm$^{-1}$ (–OH bending of structural hydroxyl), 1450 cm$^{-1}$ (–OH bending of absorbed water) and 1037 cm$^{-1}$ (Si–O stretch of silicate tetrahedral). The pattern of GO shows the bands at 3445 cm$^{-1}$ (–OH stretch), 1626 cm$^{-1}$ (–OH bending), 1384 cm$^{-1}$ (C=C bending), 1711 cm$^{-1}$ (C=O stretch of carboxyl group), 1218 cm$^{-1}$ (C–O stretch of carboxyl group) and 1045 cm$^{-1}$ (C–O–C stretch of epoxy group) [28,40,41]. In the FTIR spectrum of GO–MMT

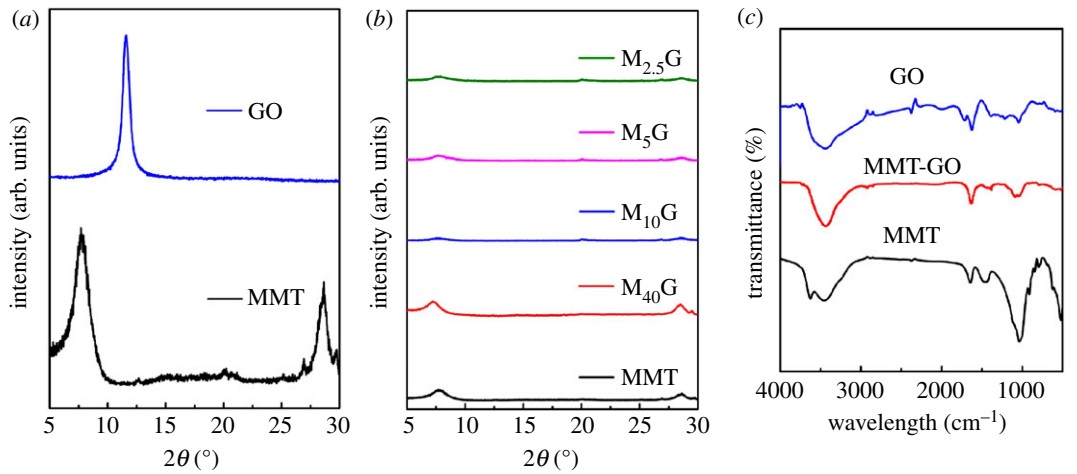

**Figure 6.** XRD patterns of GO, MMT minerals (*a*) and aerogel samples (*b*). FTIR patterns of GO, MMT minerals and M$_5$G aerogels (*c*).

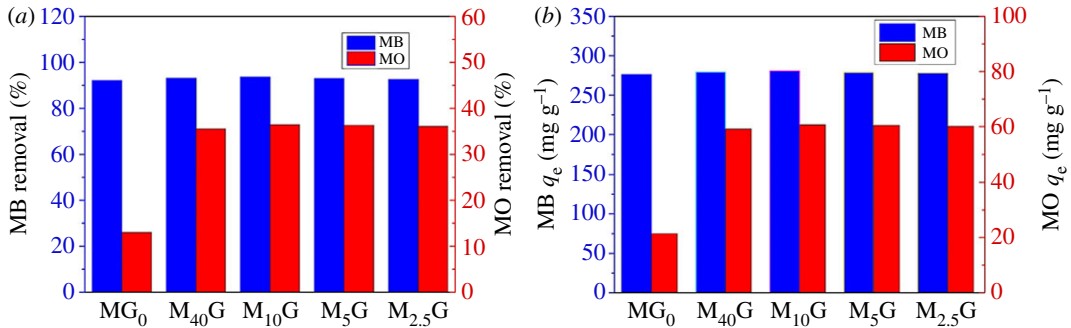

**Figure 7.** The adsorption of MB and MO onto the aerogels: (*a*) removal percentage, (*b*) adsorption capacity.

aerogel, almost all the characteristic bands of GO and MMT appear and no new bands present, indicating that there may be no chemical interaction between them in the binary sample.

GO and MMT are significant adsorbents due to their large surface area [14] and adsorption tests of cationic and anionic dyes onto the aerogels were investigated. A small amount of PVA cross-linked with glutaraldehyde was added to the aerogels to prevent the samples from falling to pieces in aqueous solution. Figure 7*a,b* show that the adsorption efficiency of MB onto the adsorbents is much higher than that of MO. This can be explained by the fact that MMT and GO nanosheets are negative charged and so they normally adsorb high amounts of cation dye due to an electrostatic attraction [42]. With the increase in GO and the decrease in MMT (from MG$_0$ to M$_{2.5}$G) in the aerogels, the adsorption capacity and removal percentage of MO rapidly increase and then keep stable, while those of MB slightly increase and then decrease, both reaching the maximum value at M$_{10}$G. It is well known that GO has plenty of oxygen atoms on the graphitic backbone in the forms of epoxy, hydroxyl and carboxyl groups [43,44]. These groups increase the binding capacity of organic dyes [45] and endow GO high adsorption efficiency, so the adsorption capacity increases with the increase in GO content. However, with the further increase in GO, the edges of two adjacent GO flakes may adhere with each other, which leads to the wrapped MMT particles between GO flakes and the decrease of the exposed surface area of MMT platelets. So, MMT–GO aerogels with a small amount of GO show good adsorption efficiency for removing organic dyes from environmental solutions.

## 4. Conclusion

In this study, GO is prepared and MMT is exfoliated into nanoparticles with two to three layers. The texture of MMT–GO binary dispersion was investigated and MMT–GO aerogels were synthesized. The introduction of MMT nanosheets enhances LCs formation of GO because MMT platelets show ordered alignment through the templating role of GO LCs. For MMT–GO gel, the highly ordered texture is

observed, and LCs evolve to more ordered mesophases with the increase in GO concentration. When the gel is freeze-dried, the aerogels with interconnected three-dimensional pores are obtained and the pore walls consist of ordered sheets of GO and MMT, which is attributed to LCs self-assembly and ice-templating role. Moreover, the aerogel samples with GO/MMT ratio of 1 : 10 show better adsorption capacity of MB and MO, reaching the maximum values of 280.8 and 60.67 mg g$^{-1}$, respectively. In view of the above, MMT and GO show the synergistic effect of LCs formation and they could be used to assemble macroscopic aerogels that are effective adsorbents for removals of the anionic and cationic dyes.

Data accessibility. The data supporting this paper have been uploaded as part of the electronic supplementary material.

Authors' contribution. S.Y. and Z.Y. conducted molecular laboratory work, participated in data analysis, performed sequence comparison, participated in research design and drafts manuscripts; Z.S. conducted statistical analysis; J.Xie collected field data; Z.Y. and J.Xu conceived the study, designed the study, coordinated, researched and helped draft the manuscript. All authors eventually approved the publication.

Competing interests. We declare we have no competing interests.

Funding. The study was funded by Macheng Tourism Investment Co., Ltd in China (grant no. KH096-340) and by Wuhan Jinaide Technology Co., Ltd in China (grant no. KH126413).

Acknowledgements. The authors gratefully acknowledge the financial support by Macheng Tourism Investment Co., Ltd in China (grant no. KH096-340) and by Wuhan Jinaide Technology Co., Ltd in China (grant no. KH126413).

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
