## [Reviewer comments · Royal Society Open Science]

Review History

RSOS-181439.R0 (Original submission)

Review form: Reviewer 1

Is the manuscript scientifically sound in its present form?

Yes

Are the interpretations and conclusions justified by the results?

Yes

Is the language acceptable?

No

Is it clear how to access all supporting data?

Yes

Do you have any ethical concerns with this paper?

No

Have you any concerns about statistical analyses in this paper?

No

Recommendation?

Accept with minor revision (please list in comments)

Comments to the Author(s)

The authors have combined graphene oxide and montmorillonite sheets using the freeze drying process for producing aerogels. As these are both plate-like substrates, it makes sense that they could align. The authors show that under ideal conditions, liquid crystallinity is observed, which is very nice!

The use of English requires a great deal of work. I recommend that the authors find a native speaker willing to heavily edit the work - there are words that I don't understand "nicols?", phrases such as "primarily prepared" and misspellings, such as "dramasticly". If the authors have access to MS Word, the spell and grammar check features should identify most of these issues. Use of "sol-gel" will be confusing to aerogel readers, who usually associate that phrase with the silica/hydrolysis route. It is unnecessary to say sol-gel. Clay forms a hydrogel, and can bond the GO.

In the conclusions, the statement about using these materials to clean sewage streams of dyestuffs is not realistic. These aerogels are very expensive and are water sensitive. There really is no chance of such a large scale application. Better to just state that the absorbance performance has been demonstrated.

Review form: Reviewer 2

Is the manuscript scientifically sound in its present form?

Yes

Are the interpretations and conclusions justified by the results?

Yes

Is the language acceptable?

No

Is it clear how to access all supporting data?

Yes

Do you have any ethical concerns with this paper?

No

Have you any concerns about statistical analyses in this paper?

No

Recommendation?

Accept with minor revision (please list in comments)

Comments to the Author(s)

The present paper mainly deals about the arrangement and self-assembly of graphene oxide (GO) and montmorillonite nanoplatelets (MMT) in water suspension used as precursor to GO/MMT aerogels. Although the matter of the paper is not fully new, it contains remarkable contributions to

the knowledge in the field justifying its publication. Nevertheless I would like to point out some minor issues that authors may address before publication.

1. Even though I am not a native English speaker, I have found some grammatical errors that are difficulting the correct understanding of the work. I suggest revising the language used.
2. FTIR section. I agree that no reaction between MMT and GO exists. However, the statement at the end of page 18, line 46, is not fully accurate. The absence of signal in the FTIR analysis may be also due to the detection limit of the technique. The absence of signal does not necessarily guarantee the absence of reaction.
3. In the investigation of the adsorption capability an aerogel is formed by adding PVA and crosslinking with glutaraldehyde. It would be good to know if the previously reported structures (figure 5) are or not maintained. In other words, it would be recommendable showing the structures of the PVA-based aerogels because these systems are in fact used in the final application (dye adsorption)

Decision letter (RSOS-181439.R0)

19-Nov-2018

Dear Dr Yang:

Title: Clay - graphene oxide Liquid crystals and their aerogels : synthesis, characterization and properties

Manuscript ID: RSOS-181439

Thank you for submitting the above manuscript to Royal Society Open Science. On behalf of the Editors and the Royal Society of Chemistry, I am pleased to inform you that your manuscript will be accepted for publication in Royal Society Open Science subject to minor revision in accordance with the referee suggestions. Please find the reviewers' comments at the end of this email.

The reviewers and handling editors have recommended publication, but also suggest some minor revisions to your manuscript. Therefore, I invite you to respond to the comments and revise your manuscript.

Please also include the following statements alongside the other end statements. As we cannot publish your manuscript without these end statements included, if you feel that a given heading is not relevant to your paper, please nevertheless include the heading and explicitly state that it is not relevant to your work. We have included a screenshot example of the end statements for reference.

- Ethics statement

Please clarify whether you received ethical approval from a local ethics committee to carry out your study. If so please include details of this, including the name of the committee that gave consent in a Research Ethics section after your main text. Please also clarify whether you received informed consent for the participants to participate in the study and state this in your Research Ethics section.

OR

Please clarify whether you obtained the necessary licences and approvals from your institutional animal ethics committee before conducting your research. Please provide details of these licences and approvals in an Animal Ethics section after your main text.

OR

Please clarify whether you obtained the appropriate permissions and licences to conduct the fieldwork detailed in your study. Please provide details of these in your methods section.

Because the schedule for publication is very tight, it is a condition of publication that you submit the revised version of your manuscript before 28-Nov-2018. Please note that the revision deadline will expire at 00.00am on this date. If you do not think you will be able to meet this date please let me know immediately.

Best wishes,
Dr Laura Smith
Publishing Editor, Journals

On behalf of the Subject Editor Professor Anthony Stace and the Associate Editor Dr Andrew Harned.

RSC Associate Editor:

Comments to the Author:

The referees have shown enthusiasm for this work, but have raised important concerns regarding the overall readability of the manuscript. I highly recommend the authors modify their manuscript with these comments in mind. Modifying their introduction and conclusions with regard to the potential applicability of these materials may also be warranted.

RSC Subject Editor:

Comments to the Author:

(There are no comments.)

Reviewer comments to Author:

Reviewer: 1

Comments to the Author(s)

The authors have combined graphene oxide and montmorillonite sheets using the freeze drying process for producing aerogels. As these are both plate-like substrates, it makes sense that they could align. The authors show that under ideal conditions, liquid crystallinity is observed, which is very nice!

The use of English requires a great deal of work. I recommend that the authors find a native speaker willing to heavily edit the work - there are words that I don't understand "nicols?", phrases such as "primarily prepared" and misspellings, such as "dramasticly". If the authors have access to MS Word, the spell and grammar check features should identify most of these issues. Use of "sol-gel" will be confusing to aerogel readers, who usually associate that phrase with the silica/hydrolysis route. It is unnecessary to say sol-gel. Clay forms a hydrogel, and can bond the GO.

In the conclusions, the statement about using these materials to clean sewage streams of dyestuffs is not realistic. These aerogels are very expensive and are water sensitive. There really is no chance of such a large scale application. Better to just state that the absorbance performance has been demonstrated.

Reviewer: 2

Comments to the Author(s)

The present paper mainly deals about the arrangement and self-assembly of graphene oxide (GO) and montmorillonite nanoplatelets (MMT) in water suspension used as precursor to GO/MMT aerogels. Although the matter of the paper is not fully new, it contains remarkable contributions to the knowledge in the field justifying its publication. Nevertheless I would like to point out some minor issues that authors may address before publication.

1. Even though I am not a native English speaker, I have found some grammatical errors that are difficult to understand. I suggest revising the language used.
2. FTIR section. I agree that no reaction between MMT and GO exists. However, the statement at the end of page 18, line 46, is not fully accurate. The absence of signal in the FTIR analysis may be also due to the detection limit of the technique. The absence of signal does not necessarily guarantee the absence of reaction.
3. In the investigation of the adsorption capability, an aerogel is formed by adding PVA and crosslinking with glutaraldehyde. It would be good to know if the previously reported structures (figure 5) are or are not maintained. In other words, it would be recommended showing the structures of the PVA-based aerogels because these systems are in fact used in the final application (dye adsorption).

Author's Response to Decision Letter for (RSOS-181439.R0)

See Appendix A.

Decision letter (RSOS-181439.R1)

14-Jan-2019

Dear Dr Yang:

Title: Clay - graphene oxide Liquid crystals and their aerogels : synthesis, characterization and properties

Manuscript ID: RSOS-181439.R1

It is a pleasure to accept your manuscript in its current form for publication in Royal Society Open Science. The chemistry content of Royal Society Open Science is published in collaboration with the Royal Society of Chemistry.

On behalf of the Subject Editor Professor Anthony Stace and the Associate Editor Dr Andrew Harned.

RSC Associate Editor
Comments to the Author:
(There are no comments.)

Reviewer(s)' Comments to Author:

Appendix A

Dear Editor,

We would like to resubmit the revised manuscript entitled “Clay-gaphene oxide Liquid crystals and their aerogels: synthesis, characterization and properties” for consideration by the Royal Society Open Science. We would like to thank the reviewers for thoroughly reviewing our manuscript and making many thoughtful comments. We have revised the manuscript extensively. And mark the modified part in red. Here are our point-by-point responses:

Reviewer #1:

Comment 1: In the conclusions, the statement about using these materials to clean sewage streams of dyestuffs is not realistic. These aerogels are very expensive and are water sensitive. There really is no chance of such a large scale application. Better to just state that the absorbance performance has been demonstrated.

Answer: This suggestion is constructive and helps us improve our manuscript. We have made changes and removed the unreasonable points. These changes are provided in line 16-18 in page 12, line 30 in page 12 and line 1-2 in page 13.

Comment 2: Use of "sol-gel" will be confusing to aerogel readers, who usually associate that phrase with the silica/hydrolysis route. It is unnecessary to say sol-gel. Clay forms a hydrogel, and can bond the GO.

Answer: To make the readers more understandable, we have changed the "sol-gel" where the aerogel readers are confused.

Reviewer #2:

Comments 1 : Even though I am not a native English speaker, I have found some grammatical errors that are difficulting the correct understanding of the work. I suggest revising the language used.

Answer: We are very sorry for our English writing. English of this manuscript has been improved, including grammar, vocabulary and sentences, etc.

Comments 2 : FTIR section, I agree that no reaction between MMT and GO exists. However, the statement at the end of page 18, line 46, is not fully accurate. The absence of signal in the FTIR analysis may be also due to the detection limit of the technique. The absence of signal does not necessarily guarantee the absence of reaction.

Answer: In FTIR section, the inaccurate description has been revised. They are changed in line 19-22 in page 11.

Comments 3: In the investigation of the absorption capability an aerogel is formed by adding PVA and crosslinking with glutaraldehyde. It would be good to know if the previously reported structures (figure 5) are or not maintained. In other words, it would be recommendable showing the structures of the PVA-based aerogels because these systems are in fact used in the final application (dye adsorption)

Answer: We added a small amount of PVA crosslinked with glutaraldehyde (GA) to the aerogel to prevent the sample from falling to pieces in aqueous solution. From SEM images below (Fig.1), we found that the addition of PVA and glutaraldehyde had little effect on the structure of GO-MMT aerogels, which is attributed to their small amounts. So, the effect of the addition of these materials on the aerogel structure was not discussed in the manuscript.

Fig.1 SEM image of GO -MMT-PVA-GA aerogel. Scale bar: 100 μm .

Once again, thank you very much for your constructive comments and suggestions which would help us in depth to improve the quality of the paper.

Sincerely yours,

Zhihong Yang

Faculty of Materials Science and Chemistry,

China University of Geosciences, Hubei, Wuhan 430074, P. R. China

E-mail: yzhh05@126.com